# Influence of Process Fluctuations on Residual Stress Evolution in Rotary Swaging of Steel Tubes

**DOI:** 10.3390/ma12060855

**Published:** 2019-03-14

**Authors:** Svetlana Ishkina, Dhia Charni, Marius Herrmann, Yang Liu, Jérémy Epp, Christian Schenck, Bernd Kuhfuss, Hans-Werner Zoch

**Affiliations:** 1Bremen Institute for Mechanical Engineering-bime, University of Bremen, Badgasteiner Str. 1, 28359 Bremen, Germany; herrmann@bime.de (M.H.); sea.yang.liu@gmail.com (Y.L.); schenck@bime.de (C.S.); kuhfuss@bime.de (B.K.); zoch@iwt-bremen.de (H.-W.Z.); 2Leibniz Institute for Materials Engineering-IWT, Badgasteiner Str. 3, 28359 Bremen, Germany; charni@iwt-bremen.de (D.C.); epp@iwt-bremen.de (J.E.); 3MAPEX Center for Materials and Processing, University of Bremen, Bibliothekstr. 1, 28359 Bremen, Germany

**Keywords:** finite element method, incremental bulk forming, cold forging, X-ray diffraction

## Abstract

Infeed rotary swaging is a cold forming production technique to reduce the diameter of axisymmetric components. The forming is achieved discontinuously by a series of radial strokes that are spread over the shell of the part. Due to tolerances within the rotary swaging machine, these strokes perform individually and the resulting stroke pattern is not homogeneous with regards to circumferential and longitudinal distribution. Nevertheless, in combination with the high number of performed strokes and the large contact area between the dies and the part, the external part properties, such as diameter, roundness and surface roughness, show even values along the finished part. In contrast, strength-defining internal part properties, like microstructure and residual stress components, are more sensitive to the actual pattern and temporal sequence of the individual strokes, which is investigated in this study. The impact of process fluctuations during the conventional process, which are induced by the tolerances of machine tool components, was verified by numerical simulations, physical tests and measurements of residual stress distributions at the surface and at depth. Furthermore, a method is introduced to maintain the stroke following angle ∆φ at zero by flat dies, and thus the actual pattern and temporal sequence of the strokes was homogenized. The results show that the residual stress fluctuations at the surface could be controlled and reduced. Furthermore, it is demonstrated that the depth profile of the residual stresses at a distance of 300 µm from the surface developed independently from the process fluctuations.

## 1. Introduction

Rotary swaging is an incremental cold forming process for reducing and profiling the cross section of axisymmetric components, such as bars and tubes. It is established in the automotive and aerospace manufacturing industries [1]. This production technique provides several advantages such as high static and dynamic strength due to strain hardening [2]. Additionally, by the use of hollow shafts and by the advantageous development of the wall thickness during rotary swaging, cost-effective production of lightweight components is possible due to the optimal use of material resources [3].

In infeed rotary swaging, the part is axially fed into the swaging head with the feed velocity v_f_ (see Figure 1). The diameter is reduced incrementally by a revolving and radially oscillating motion of (four) dies. This motion is induced by the rotating swaging axle that entrains the base jaws with cams on the top surface. These cams pass the cylinder rollers and push the tools inwards. The swaging head exhibits 12 cylinder rollers. The oscillation of the dies is initiated and controlled by the dimensions and proportions of machine tool components in the swaging head that constitute the pressure column between the part and outer ring of the machine frame. These components include the dies, the spacers, the base jaws with cams and the cylinder rollers. All of these machine parts feature technical tolerances. Hence, the geometrical deviation of the pressure columns that reassemble with every single stroke generate a scatter within the impact pattern on the surface of the part.

During the closing of the swaging dies, the part may be shifted backwards due to the die angle in the reduction zone that is too steep to ensure self-locking. When the plastic deformation starts, the material flows mainly in the feed direction (positive material flow), but also in the negative direction [4]. The resulting sum of the backward shift of the part and the negative material flow constitutes the axial back pushing. This back pushing can be measured during the process [5]. It is strictly related to the actual tribological conditions in the contact area between the part and dies. Consequently, the impact pattern is further distorted. Despite this inhomogeneity and the discontinuous nature of the process, the combination of the axial feed per stroke value l_st_ of the part and the circumferential shift of the dies between consecutive strokes allows for producing parts with constant external properties, like final diameter, length, roundness and surface roughness, over the entire formed length [6]. During the process, every spot of the finished part experiences about 100 effective strokes.

Several studies have been carried out to establish optimal process conditions concerning surface quality and geometry, e.g., by analyzing the influence of dry process conditions [5], feed per stroke values [6], radial feed per stroke values during plunge rotary swaging [7] or true strain [8]. Furthermore, the internal properties of the part, such as its microstructure, hardness and residual stress components, which are important for its strength, have been investigated: Lim found that the hardness at the outer surface increased with increasing diameter reduction of steel tubes [9] and Kocich stated that rotary swaging induces a strong fiber texture in tungsten heavy alloys [10]. Simulations are used to investigate the process and the material behavior during rotary swaging. For example, the material flow [4], plastic strain [11] and strain rates [12] have been investigated. The von Mises equivalent stress was examined by Ameli et al., who found it to be directly related to the residual stresses [13]. They found that the residual stresses were mostly affected by the feed per stroke. Furthermore, Ghaei found, in his finite element rotary swaging simulation, that the axial stresses of steel tubes could be controlled by the die geometry. Additionally, he showed that tubes featured higher tensile residual stress at the outer surface after rotary swaging without mandrel compared to rotary swaging with mandrel, in both the axial and tangential direction [14]. In general, the residual stresses are strongly related to the plastic strain during rotary swaging and exhibit much larger scattering than the produced geometry. In [15], the evolution of the surface residual stress after rotary swaging reflected the great fluctuations of lubricated as well as dry formed parts. The investigations showed that this inhomogeneity occurred due to the complexity of both the interaction between dies and part as well as the material flow history. The results were also influenced by the feed velocity v_f_. Another study of the swaged tubes showed that the strain was very high at the outer surface of the tube and decreased strongly towards the inner wall of the tube [16]. Furthermore, rotary swaging could introduce material inhomogeneity, which could be influenced by the process parameters [17].

The main question in this work is whether intrinsic process fluctuations that are generated by technical tolerances of the machine components influence the scattering of the resulting part properties, in particular, the imprinted residual stress components. The employed method to analyze the influence of spreading of the strokes on the part surface was to set the stroke following angle ∆φ = 0°. In this case, the part rotates with the same angular velocity as the swaging axle. In consequence, the same pressure column assemblies hit the same side of the part in a short sequence [18]. Simulations of the rotary swaging process with constant and with varying stroke heights were carried out. In experimental tests, the actual process parameters were monitored during rotary swaging of tubes and the resulting residual stresses of the produced parts were measured afterwards.

## 2. Methods

### 2.1. Modelling and Simulation

For this study, a 2D axisymmetric model was built in the general finite element software ABAQUS (version 6.14-1, Dassault Systèmes, Vélizy-Villacoublay, France). 2D models are often used due to their short calculation time compared to 3D models. The chosen assumptions were: the material model was based on combined hardening (kinematic and isotropic); the friction coefficient was based on coulomb’s law [19] (lubricated contact pair steel-steel µ = 0.1 [20]) with penalty friction formulation and was assumed to be constant over the complete contact area and during the whole forming; the process was adiabatic, thus the thermal issues including plastic heat generation, heat conduction and heat exchange were neglected; the die was modelled as rigid and the tube as a deformable body. The plastic properties of the part (steel E355) were experimentally determined by tensile tests. Further parameters were d_0_ = 20 mm, s_0_ = 3 mm, d_1_ = 15 mm (see Figure 2). In the calculation, a CAX4RT mesh was applied to the part. The mesh size was set to 0.1 mm in the radial direction and to one tenth of the feed per stroke value l_st_ in the axial direction. The feed velocity was set to v_f_ = 2000 mm/min (l_st_ = 0.92 mm). The feeding of the part as well as the stroke of the dies were controlled by a time series (displacement tables). The motion of the dies was designed by the shape of the cam and the diameter of the cylinder rollers. The stroke height (h_T_) was principally set to 1 mm. To ensure that the material experienced the whole deformation process, the feed length (z) was set to 100 mm. At the end of the dynamic explicit analysis, the dies were removed and the simulation was proceeded by a static general step to release the springback [21] in order to obtain the residual stress components.

To investigate the influence of small changes in the process, the simulation was performed with fluctuating die strokes (SIM_Stroke_), as well as without fluctuations (SIM_Ideal_). A change in the stroke height h_T_ was realized in such a way that every third stroke was decreased by 10 µm, which was the tolerance of the cylinder roller diameter of the rotary swaging machine that was used for experiments. The axial stress component S_zz_ was analyzed at the outer surface of the tubes in the region z_analyzed_ = 28 mm between the points z_1_ to z_2_ without load.

### 2.2. Rotary Swaging Setup

Rotary swaging experiments were carried out with E355 steel tubes (Mannesmann precision tubes, Helmond, Netherlands) with an initial length of l_0_ = 300 ± 0.85 mm, an initial diameter of d_0_ = 20 ± 0.30 mm and an initial wall thickness of s_0_ = 3 mm (as specified by the manufacturer). These tubes were produced by cold drawing. To eliminate existing residual stresses and ensure a homogeneous microstructure, a normalization heat treatment was carried out in a vacuum furnace before rotary swaging. The tubes were purged with Nitrogen gas, N_2_, (2 bars) in a vacuum furnace (Ipsen, Kleve, Germany) before establishing the vacuum environment. The samples were then gradually heated to 890 °C and maintained at this temperature for 5 h. After that, they were cooled to 600 °C at 1.5 K/min for 2 h, then to 220 °C at 5 K/min, and finally slowly cooled to room temperature. The yield strength of the heat treated material was 372 MPa and the tensile strength was 536 MPa. Round dies, see Figure 3a, made by powder-metallurgical ASP^®^2023 steel were used for the swaging process. The die angle of the reduction zone was α = 10° and the length of the calibration zone was l_cal_ = 20 mm. The reduction zone featured a spray-coated tungsten carbide layer to increase the effective friction between the dies and the part in order to reduce the axial reaction force during the process.

The experiments with the stroke following angle ∆φ = 0° were realized by using flat swaging dies, see Figure 3b, made of 1.2379 steel. These dies featured a flat surface in the reduction as well as in the calibration zones. The die angle was also α = 10°. The closed dies formed a square with a dimeter of the inner circle equal to 15 mm. All experiments were carried out with lubrication, using mineral oil Condocut KNR 22. During swaging, the parts were fixed in a hydraulic clamping device. After axial feeding of z = 130 mm into the swaging unit, the swaging axle rotation was stopped to prevent additional strokes from affecting the surface and the part was pulled out of the swaging head. The rotary swaging experiments were performed to reduce the initial diameter d_0_ = 20 mm to a final diameter/edge length of d_1_ = 15 mm. In addition to the die geometry, the feed velocity was varied. The main experimental settings are given in Table 1. Only one parameter was changed in each experiment. For each set of parameters, five samples were swaged. During the process, the rotation of the tubes formed by round dies was recorded from the first contact of the tubes with the dies until the deformation was completed.

### 2.3. Residual Stress Measurement

Residual stress distributions near the surface are important for the fatigue life [22]. The residual stresses were therefore extensively measured at different samples by X-Ray diffraction (XRD). The measurements were performed using a GE inspection technologies (GE inspection technologies, Ahrensburg, Germany) diffractometer Type ETA 3003 equipped with a 2D-detector. A Vanadium filtered Cr-Kα radiation was used with a beam diameter of 1 mm. The resulting {211} diffraction peak of α-iron was evaluated using the sin^2^ψ-method [23]. Axial and tangential residual stresses were measured along a path of 40 mm on the outer surface of the tubes with a step size of 1 mm starting 30 mm away from the calibration zone (see Figure 4). To obtain additional information on the microstructure, the full width at half maximum (FWHM) of the diffraction peaks was also evaluated. In cold forming applications of non-hardened steel, the FWHM is strongly influenced by the cold working and correlates directly with the hardness (high FWHM value reflects high hardness), which allows a qualitative evaluation of its distribution [24]. The residual stresses and the FWHM of the diffraction peaks were measured on the selected samples from each test run. In addition to the surface measurements, analyses of residual stresses and FWHM were also performed in depth. For this, the material removal was done by electrochemical etching using a solution of H_3_PO_4_ (phosphoric acid) and H_2_SO_4_ (sulfuric acid) with a volume ratio of 4:1.

## 3. Results

### 3.1. Simulation

The simulation results reflect the behavior of the material flow during the process. The ratio of kinetic and internal energy gained a maximum value of 4.5% at the very first contact of the part and die, which is lower than the generally accepted limit of 10%. The axial stress component S_zz_, which accords with the axial residual stress component along the outer surface of the tube, is shown in Figure 5. In the SIM_Ideal_ with constant feed per stroke and constant stroke height values, S_zz_ at the outer surface fluctuated only marginally. However, small changes in the stroke height of 10 µm already led to valley peaks in the axial stress component S_zz_ distribution with a difference of about 20% (see SIM_Stroke_).

Residual stress components are sensitive to the process parameters and their fluctuations. Such fluctuations are likely in rotary swaging due to the diameter tolerance of the cylinder rollers of ± 5 µm and the tolerances of the initial part geometry. Additionally, vibrations of the machine can further disturb the stroke height during rotary swaging.

### 3.2. Process Analysis

By using flat dies, the stroke following angle ∆φ was forced to be 0°, hence the sample rotated with the same velocity as the swaging axle and the tools. In this case, the rotation N over the complete forming processes was equal to 78.8 revolutions for v_f_ = 500 mm/min, 39.4 revolutions for v_f_ = 1000 mm/min and 19.7 revolutions for v_f_ = 2000 mm/min. The mean value and the deviation of the part rotation by swaging with round dies are illustrated in Figure 6. This rotation of the samples in spite of its fixation in the feeding device was caused by the rotating dies. The part rotated less with higher feed per stroke values l_st_ due to the reduced number of strokes that acted on the tube. A regression line for the part rotation can be calculated by Equation (1):
(1)$$N=\frac{z\times \phi_{part}}{l_{st}\times 360{}^{\circ}}$$
The angle φ_part_ is the part rotation, which is induced by closed swaging dies during the rotation of the axle. However, the value of the part rotation deviated from the reference line shown in Figure 6. This can be explained by the contact time of the part and the dies. With higher feed per stroke values, the part is fed deeper into the dies between consecutive strokes and φ_part_ is higher due to longer contact during the process.

### 3.3. Residual Stress Analysis

For most samples that were rotary swaged by round dies, a fluctuating compressive stress was observed along the axial position with few points in the tensile stress area. Residual stress measurements of the tube sample 1 of R_0500_ showed a periodic fluctuation in both axial and tangential directions, see Figure 7. The FWHM exhibited small variations, but with nearly the same periodicity as the residual stress values. Most of the samples processed with round dies showed comparable distributions and all samples exhibited similar residual stress tendencies in both axial and tangential directions. Therefore, only axial residual stresses were further considered in this work.

The material in its initial state was exposed to residual stress relaxation heat treatment, thus it can be assumed that the observed residual stress fluctuations result from the rotary swaging process as postulated in the previous section. Furthermore, additional experiments were performed using dies with a diamond-like carbon (DLC) coating [25] and without the rough tungsten carbide coating. The DLC coating had a thickness of 2.2 μm and an indentation hardness H_IT_ of 18 GPa. Residual stress and FWHM were evaluated and comparable fluctuations were also observed at the surface. Hence, that coating was not the cause of these fluctuations.

To further investigate these fluctuations, measurements were carried out at several positions below the surface of a different R_0500_ sample, sample 2, which showed similar fluctuating residual stresses as observed in the R_0500_ sample 1 with different magnitudes (see Figure 8a). In particular, a highly pronounced residual stress peak between the axial positions of 20 mm and 30 mm shifted the residual stress values to tensile. This is however not a singular effect, since the residual stresses showed periodic distribution with variable amplitude over the entire analyzed region. This periodicity was observed in all samples processed with this strategy (see Figure 7, Figure 8 and Figure 9) and was induced by the fluctuation of local deformation during the process. This means that the fluctuations are a signature of the rotary swaging process in the employed setup. Within few tens of micrometers below the surface, the fluctuation of residual stress values were still persistant but progressively decreased and the values shifted to tension. In the same way, the FWHM (see Figure 8b) decreased at 10 µm below the surface and the variations were also progressively reduced between 10 µm and 40 µm in depth.

Additionally, several samples showed strong scattering and no clear trend of the residual stresses like both samples of condition R_2000_, which were produced with exactly the same process parameters (compare sample 1 and sample 2 in Figure 9). On the one hand, some of the measured points of both samples exhibited tensile stresses (from 10 mm to 17 mm) and showed large differences between both samples of 276 MPa to −396 MPa. On the other hand, some positions showed comparable values of compressive stresses for both samples (from 30 mm to 40 mm). From these results it can be stated that the process reproducibility in terms of residual stress generation is not given. This low reproducibility of the internal properties could be caused not only by the individuality of the strokes, which is also shown by the simulation results with the change of the stroke height values (SIM_Stroke_), but also possibly by the sensitvity of the process to external factors, such as the initial sample axial deviation or tube wall eccentricity that are common issues in mandrel-free rotary swaging. In addition, the sampling resolution of the X-ray diffraction method (XRD) was too low (about 1 mm) to reflect changes of the residual stresses in the range of the analyzed feed per stroke values (l_st_ < 1 mm). Thus, the fluctuations with respect to the fluctuations of the process could be seen only conditionally.

In order to determine the cause of the residual stress fluctuations, process variations using flat dies were performed. Contrarily to the round bars from the previous investigations, the samples produced using the flat die setup exhibited square sections due to ∆φ = 0°, with quite homogenous distributions of residual stresses at the outer surface, with values between 400 MPa and 500 MPa tensile stress (see Figure 10). This homogeneity could also be observed by the FWHM distribution with small variations around 2.35°. For the higher feeding velocity (see Figure 10b), slightly higher variations resulted but they were still low compared to the case of using round dies and arbitrary ∆φ.

From the previous results, it could be observed that rotary swaging using flat dies and a stroke following angle ∆φ = 0° led to much higher homogeneity of surface residual stresses than with the previous round die setup. Additionally, the reproducibility of the process was evaluated by investigating three samples processed with the same process parameters as with the flat dies. The axial residual stress values of three samples of the experiment F_0500_ are shown in Figure 11. It can be seen that the residual stresses were all in a close range around 450 MPa to 600 MPa, while the differences in-between one sample was very low and did not exceed 100 MPa.

Complementary to the surface measurements, residual stress depth profiles were measured at different square section workpieces and are presented in Figure 12. The residual stresses in both directions, and the FWHM, changed significantly in the first 30 µm under the surface and then stabilized for deeper positions. This correlates well with residual stress depth profiles at round samples and metallographic examinations in a previous work [15] that showed a layer of highly elongated grains at the surface within a layer of 30 µm thickness. Comparable microstructure was also found for the round and the flat rotary swaging setup in the present study, see Figure 13.

A final comparison of axial residual stresses resulting from the different process variations investigated in this study is shown in Figure 14 for the surface values and the value 300 µm below the surface. The samples that were cold formed with the round die setup predominantly showed compressive axial residual stresses with high scattering, while the samples that were cold formed with the flat die setup featured homogenous tensile axial residual stresses at the surface. The average of axial residual stresses was about 300 MPa for all samples 300 µm below the surface. Based on these results, it seems that in the investigated parameter range, the residual stress distribution at surface regions is strongly influenced by the process parameter, while the evolution at depth is mostly independent. These results support the conclusions of previous investigations on S235 steel, where a similar observation was made [15].

## 4. Conclusions

In this study, infeed rotary swaging experiments with E355 steel tubes were carried out. The influence of slightly fluctuating process parameters on residual stress distribution was investigated. From the results, the following conclusions can be drawn:In simulations, the axial stress component S_zz_ depends on the individual stroke during rotary swaging. Fluctuations in the stroke height of a few microns, which are in the range of the tolerance of the components of the swaging head, lead to strong fluctuations in the simulated residual stress distribution at the surface.The recorded part rotation N over the complete forming process depends on the set parameter feed per stroke l_st_. N decreases with a decreasing number of strokes (higher feed velocity v_f_) but increases with an increased feed per stroke value due to the longer contact.The residual stress distribution at the surface and in the first few tens of microns is influenced by each stroke in the process. The fluctuations observed after swaging with round dies (arbitrary ∆φ) can be explained by the slight differences between the round and flat dies due to tolerances of the machine components, as seen in the simulation. It can be assumed, that the local plastic deformation history resulting from the series of individually performed strokes produces local individual residual stress values, at least at the very surface.The residual stresses at the surface of the round parts (arbitrary ∆φ) show strong fluctuations both in the positive and negative region. They shift with variations in tensile stress under the surface and tend to become homogeneous.At the surface, the fluctuation can significantly be reduced at the square parts by controlling the stroke following angle ∆φ to be zero. This measure leads experimentally to a constant stroke pattern that consists of a sequence of periodic impacts. The use of flat dies with a stroke following angle of 0° leads to homogeneous and reproducible residual stresses at the surface.A comparison of the global results show that the process parameters (die, stroke following angle and feeding velocity) strongly influence the surface residual stress distribution, while the evolution in depth (300 µm) is almost unchanged for all investigated process variations within this study.

The findings confirm the demand of controlling the process fluctuations in rotary swaging of tubes if beneath geometric features, also internal material properties like residual stresses are pursued. These fluctuations are unavoidable and cumulate in a stroke pattern scattering over the surface of the part. In future work, this scattering should be addressed by controlling the stroke following angle ∆φ and the feed per stroke value l_st_. Controlling ∆φ is restricted to demands concerning the part geometry. Setting the stroke following angle to zero was a convenient method to demonstrate the feasibility of controlling the residual stresses, but is not suitable for a common rotary swaging process. To yield parts with compressive residual stress and good geometrical properties like roundness, strategies must be developed to satisfy both gains of internal and external part property demands. Hence, the evaluation of residual stress distribution in the complete volume are currently ongoing using neutron diffraction to evaluate the effect of process parameters at a larger scale.

## Figures and Tables

**Figure 1 materials-12-00855-f001:**
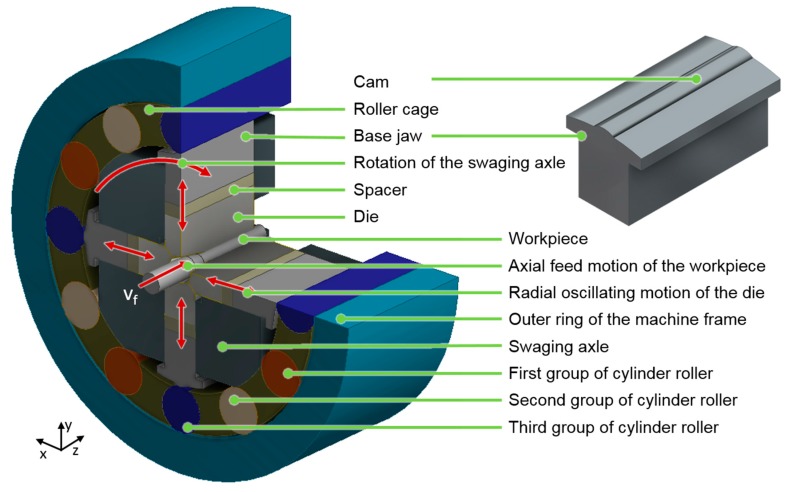
Illustration of the swaging head for infeed rotary swaging.

**Figure 2 materials-12-00855-f002:**
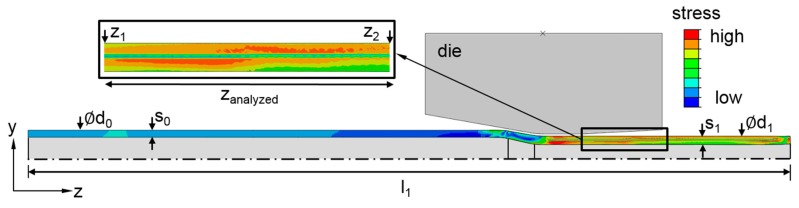
Principle of the 2D axisymmetric model with the qualitative stress result of SIM_Ideal_.

**Figure 3 materials-12-00855-f003:**
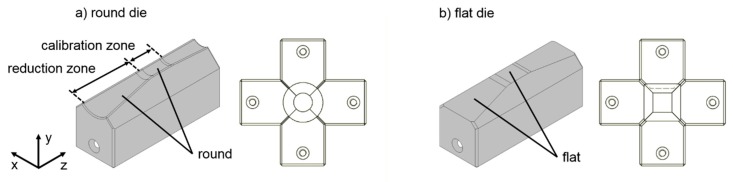
Die geometry and die assembly of the closed (**a**) round dies; (**b**) flat dies.

**Figure 4 materials-12-00855-f004:**
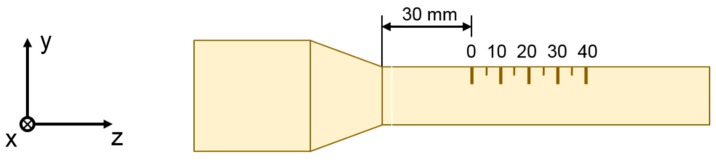
Sketch of the residual stress measurement positions (mm) at the outer surface of swaged tubes (z—axial residual stresses; x—tangential residual stresses).

**Figure 5 materials-12-00855-f005:**
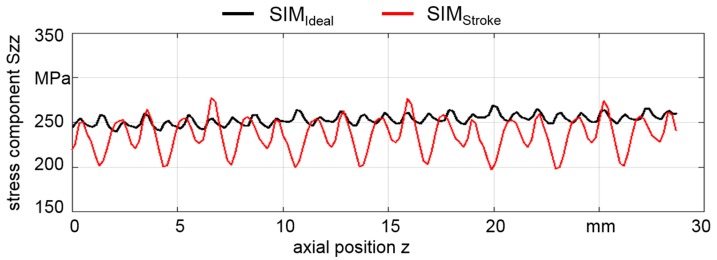
Distribution of axial stress component S_zz_ calculated by FEM on the outer surface of the tube from z_1_ to z_2_ without (SIM_Ideal_) and with (SIM_Stroke_) variations of the stroke height values.

**Figure 6 materials-12-00855-f006:**
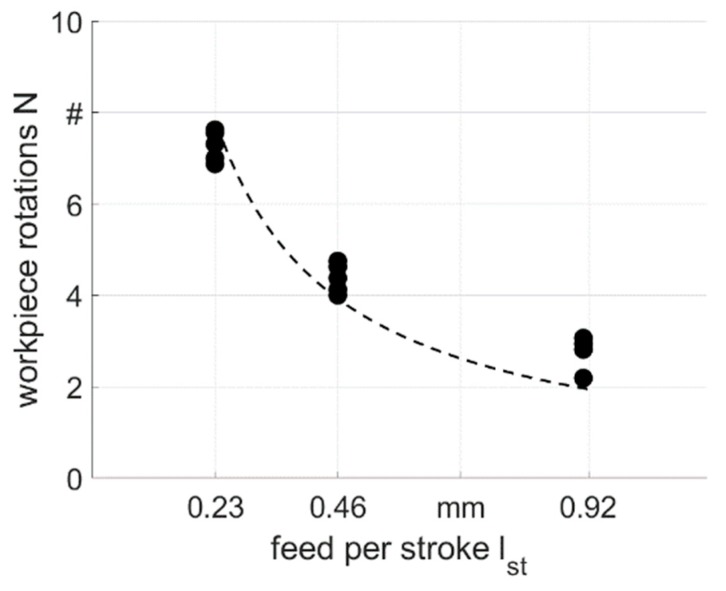
Influence of the feed velocity v_f_ on the part rotation N over the complete forming processes R_0500_, R_1000_ and R_2000_.

**Figure 7 materials-12-00855-f007:**
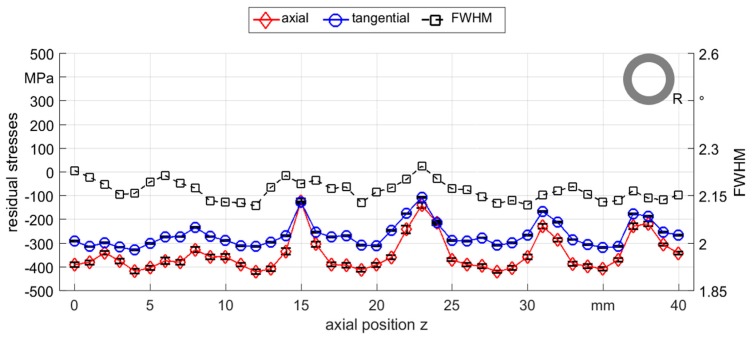
Axial and tangential residual stresses and FWHM measured along the outer surface of R_0500_ sample 1 (the full width at half maximum, FWHM).

**Figure 8 materials-12-00855-f008:**
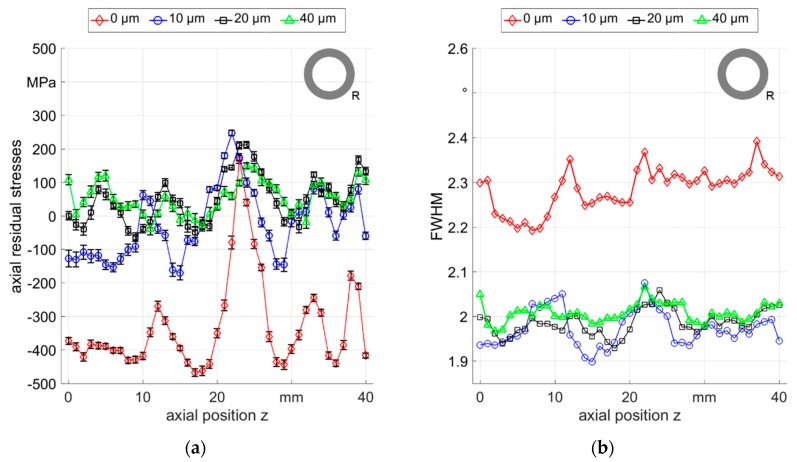
Distributions of the axial residual stresses and FWHM along 40 mm axial paths in different depths from the outer surface of R_0500_ sample 2: (**a**) axial residual stresses; (**b**) FWHM.

**Figure 9 materials-12-00855-f009:**
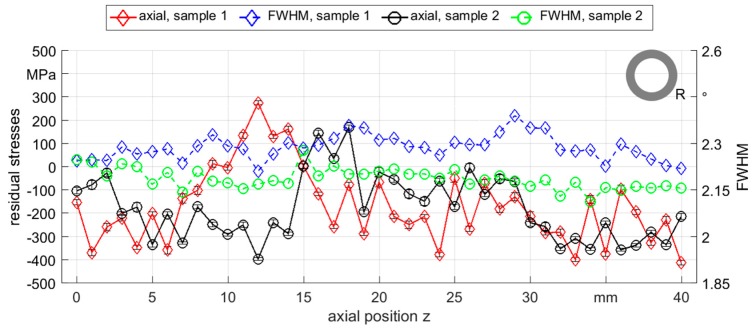
Axial residual stresses and FWHM distribution measured along the outer surface for two samples (sample 1 and sample 2) of R_2000_.

**Figure 10 materials-12-00855-f010:**
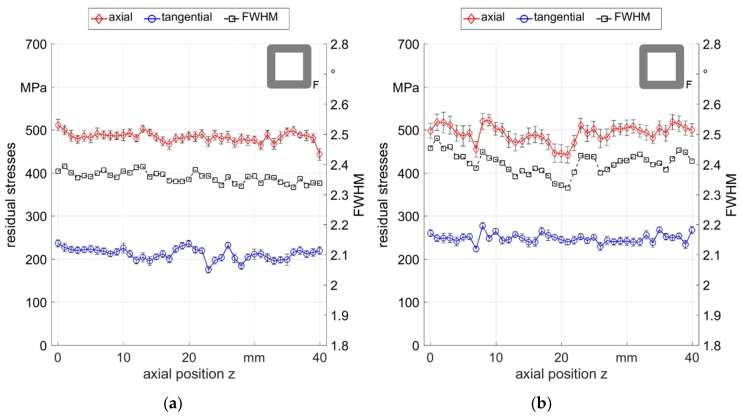
Axial and tangential residual stresses and FWHM distribution measured along the surface: (**a**) F_0500_ sample 2; (**b**) F_2000_ sample 1.

**Figure 11 materials-12-00855-f011:**
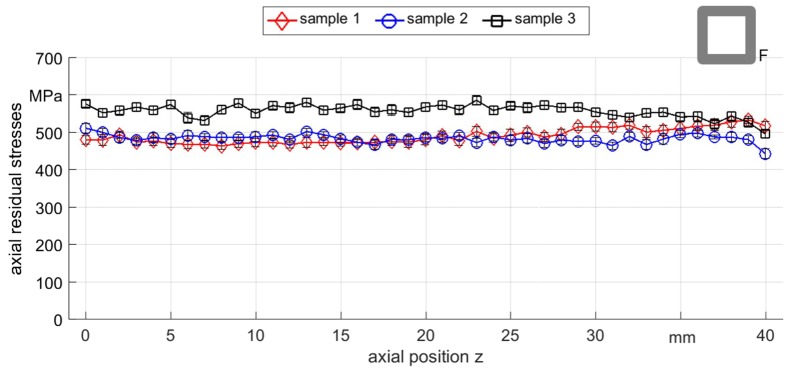
Axial residual stress distribution measured along the surface of three different samples of F_0500_.

**Figure 12 materials-12-00855-f012:**
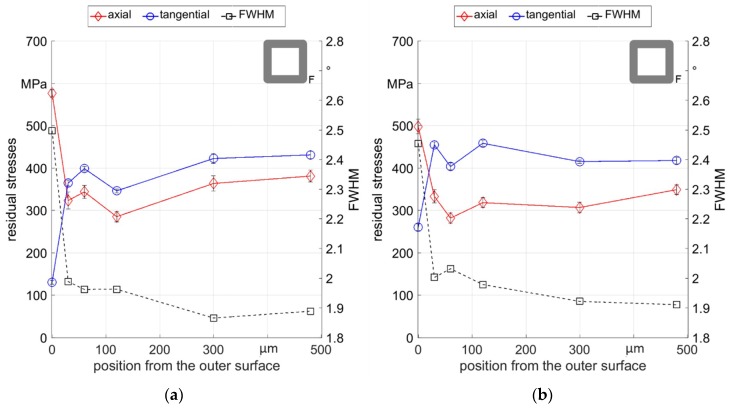
Residual stresses and FWHM depth profiles of square section samples: (**a**) F_0500_ sample 3; (**b**) F_2000_ sample 1.

**Figure 13 materials-12-00855-f013:**
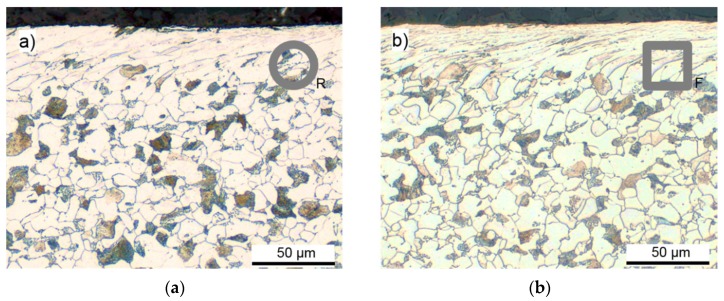
Microstructure of the swaged parts with: (**a**) round dies with arbitrary ∆φ; (**b**) flat dies with ∆φ = 0°.

**Figure 14 materials-12-00855-f014:**
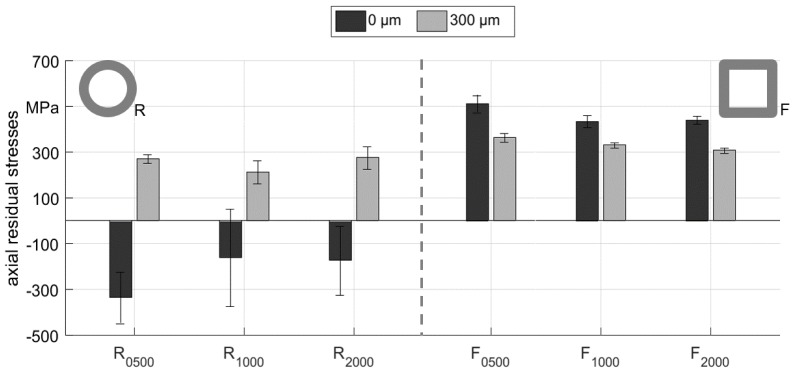
Average axial residual stress values at the surface and 300 µm below for different experiments.

**Table 1 materials-12-00855-t001:** Design of experiments.

Experiment	Feed Velocity v_f_ (mm/min)	Die Geometry	Feed per Stroke l_st_ (mm)	Stroke Following Angle ∆φ
R_0500_	500	round	0.23	arbitrary
R_1000_	1000	round	0.46	arbitrary
R_2000_	2000	round	0.92	arbitrary
F_0500_	500	flat	0.23	0°
F_1000_	1000	flat	0.46	0°
F_2000_	2000	flat	0.92	0°

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
