# Peer review of "Influence of Process Fluctuations on Residual Stress Evolution in Rotary Swaging of Steel Tubes"

_materials, 2019, doi:10.3390/ma12060855_

Reviewer 1 Report
Interesting work, however the fllowing questions arose for me:
1) It is not clear what is the material property (material model) used?
2) Also how the unloading of the material was modeled? In other words, how the material model will take into account the unloading?
3) It is not clear how the experiments for residual stress is compared with FEA?
4) How the friction coeffcienet value obtained? Is it possible it can be different for the tube during different deformation steps (because of change in surface roughness and hardening)?
5) In FAE how the loading applied? By velocity-time or acceptation time?
6) How the internal energy and dKinematic energy curves looks like?
Author Response
Dear Reviewer,
Thank You very much for your suggestions and questions to our paper. These help to improve the manuscript. We are very glad that you found our work interesting. Following changes were done:
1) It is not clear what is the material property (material model) used?
Line 106-107: “The plastic properties of the part (steel E355) were experimentally determined by tensile tests.”
Line 134-135: “The yield strength of the heat treated material is 372 MPa and the tensile strength is 536MPa.”
2) Also how the unloading of the material was modeled? In other words, how the material model will take into account the unloading?
Lines 114-116: “At the end of the dynamic explicit analysis, the dies were removed and the simulation was proceeded by a static general step to release the springback [21] to obtain the residual stress components.”
3) It is not clear how the experiments for residual stress is compared with FEA?
Line: 250-257: “This low reproducibility of the internal properties could be caused not only by the individuality of the strokes, which is also shown at the simulation results with the change of the stroke height values (SIMStroke), but also possibly by the sensitvity of the process to external factors such as the initial sample axial deviation or tube wall eccentricity that are common issues in mandrel-free rotary swaging. In adition, the sampling resolution of X-Ray diffraction method (XRD) was too low (about 1 mm) to reflect changes of the residual stresses in the range of the analyzed feed per stroke values (lst < 1 mm). Thus, the fluctuations with respect to the fluctuations of the process could be seen only conditionally.”
4) How the friction coeffcienet value obtained? Is it possible it can be different for the tube during different deformation steps (because of change in surface roughness and hardening)?
Line 102-104: “the friction coefficient was based on coulomb’s law [19] (lubricated contact pair steel - steel µ = 0.1 [20]) with penalty friction formulation and is assumed to be constant over the complete contact area and during the whole forming”
5) In FAE how the loading applied? By velocity-time or acceptation time?
Line 110-112: “The feeding of the part as well as the stroke of the dies were controlled by time series (displacement tables). The motion of the dies was designed by the shape of the cam and the diameter of the cylinder rollers.”
6) How the internal energy and Kinematic energy curves looks like?
Lines 178-180: “The ratio of kinetic and internal energy gained a maximum value of 4.5% at the very first contact of part and die that is lower than the generally accepted limit of 10%.”
Reviewer 2 Report
Dear Authors,
Congratulations on your work, which reveals a very good level of care in its presentation.
However, I have some concerns and suggestions for improvements to your work, which can be seen below:
The abstract is not clear. The ideas should be described in a clearer manner;
Figure 1 is not clear. The mechanism able to promote the oscillating motion seems to be based on different groups of rollers. However, it is not clear how the cam works;
The introduction is very well done. Congratulations!
In 2.2, I think it would be great whether the Authors can provide information about the main mechanical properties of the E355 steel;
In line 124, you refer a slow cooling. Could you provide the cooling rate value?
The text and figure from line 157 to 163 should be better explained. The draw is not clear: there is a certain distance from the tip of the sample to the first observation?
In lines 209 to 212, some details are missing about the DLC coating, namely its thickness and hardness;
;In Figure 8, there is consistent peaks of stress between 20 and 30 mm. Could you comment this trend?
In the legend of the Figure 8, please insert a blank space between the values and the units (0 um);
In Figure 9 there is a contraditory result for 12 mm, between black and red lines. Could you comment the reasons behind this behaviour?
In the conclusions, there is no reference to flat and round shape tubes. However, the results are very different, as depicted in Figure 13. Could you improve this?
Hope these suggestion help you improving your paper.
Author Response
Dear Reviewer,
Thank You very much for your suggestions to our paper. These help to improve the manuscript. We are very glad that you like our introduction. For the further notes following changes were done:
1) The abstract is not clear. The ideas should be described in a clearer manner;
Line 13-30.
2) Figure 1 is not clear. The mechanism able to promote the oscillating motion seems to be based on different groups of rollers. However, it is not clear how the cam works;
Line 43-44: “This motion is induced by the rotating swaging axle that entrains the base jaws with a cam on the top surface. This cams pass the cylinder rollers and push the tools inwards.”, Figure 1.
3) The introduction is very well done. Congratulations!
Thank You very much
4) In 2.2, I think it would be great whether the Authors can provide information about the main mechanical properties of the E355 steel;
Line 134-135: “The yield strength of the heat treated material is 372 MPa and the tensile strength is 536MPa.”
5) In line 124, you refer a slow cooling. Could you provide the cooling rate value?
Line 132-134: “After that, they were cooled to 600° at 1.5 K/min for 2 hours, then to 220° at 5 K/min, and finally slowly cooled to room temperature.”
6) The text and figure from line 157 to 163 should be better explained. The draw is not clear: there is a certain distance from the tip of the sample to the first observation?
Line 162-165: “Axial and tangential residual stresses were measured along a path of 40 mm on the outer surface of the tubes with a step size of 1 mm starting 30 mm away from the calibration zone (Fig. 4). To obtain additional information on the microstructure, the full width at half maximum (FWHM) of the diffraction peaks was also evaluated.”
Figure 4. Line 174-175
7) In lines 209 to 212, some details are missing about the DLC coating, namely its thickness and hardness;
Line 220-222: “Furthermore, additional experiments were performed using 220 dies with a Diamond Like Carbon coating (DLC) [25] and without the rough tungsten carbide coating. 221 The DLC-coating had a thickness of 2.2 μm and an indentation hardness HIT of 18 GPa.”
8) In Figure 8, there is consistent peaks of stress between 20 and 30 mm. Could you comment this trend?
Line 228-237: “To further investigate this fluctuations, measurements were carried out at several positions 228 below the surface of a different R0500 sample, sample 2, which showed similar fluctuating residual 229 stresses as observed at R0500 sample 1 with different magnitudes, see Fig. 8 (a). In particular, a highly 230 pronounced residual stress peak between the axial positions of 20 mm and 30 mm shifted the residual 231 stress values to tensile. This is however not a singular effect, since the residual stresses showed 232 periodic distribution with variable amplitude over the entire analyzed region. This periodicity 233 observed in all samples processed with this strategy (Fig. 7 to Fig. 9) was induced by the fluctuation 234 of local deformation during the process. This means that the fluctuations are a signature of the rotary 235 swaging process in the employed setup. Within few tens of micrometers below the surface the 236 fluctuation of residual stress values were still persitant but progressively decreased and the values 237 shifted to tension.”
9) In the legend of the Figure 8, please insert a blank space between the values and the units (0 um);
Done. Figure 8.
10) In Figure 9 there is a contraditory result for 12 mm, between black and red lines. Could you comment the reasons behind this behaviour?
Line 246-249: “compare sample 1 and sample 2 in Fig 9. On the one hand, some of the measured points 246 of both samples exhibit tensile stresses (from 10 mm to 17 mm) and show large differences between 247 both samples of 276 MPa to -396 MPa. On the other hand, some positions show comparable values of 248 compressive stresses for both samples (from 30 mm to 40 mm).”
11) In the conclusions, there is no reference to flat and round shape tubes. However, the results are very different, as depicted in Figure 13. Could you improve this?
Line 324, 327
Reviewer 3 Report
Review of the paper "Influence of process fluctuations on residual stress 3 evolution in rotary swaging of steel tubes" by Ishkina et al.
This paper investigated the process of infeed rotary swaging on the cold forming production technique to reduce the diameter of 14 axisymmetric components.
The ovearll papers is OK.
I advice the acceptance with minor spealing english correction.
Author Response
Dear Reviewer,
Thank You very much for your review of our paper and you suggestion. We have red the manuscript carefully and improroved the spealing mistakes.
Reviewer 4 Report
The paper reported residual stress differences based on process fluctuations. Both simulation and experimental study were conducted. It was concluded that the process should be controlled closely to minimize residual stress fluctuations. The following needs to be addressed before publication.
Please avoid grouped citation such as [5-9]. Please write one/two sentences describing their contribution to the literature.
Please mark clearly in figure 1: which is the axial direction and which is the tangential direction.
Page 9: line 264: Comparable microstructure was also found for the round and the flat rotary swaging setup in the present study. Please provide evidence for this statement.
Figure 7 and 8: Why the axial residual stress at the surface in figure 8 is different than the same positions in figure 7? It is a very crucial point.
Figure 12: What is the depth of the residual stress measurement points?
Author Response
Dear Reviewer,
Thank You very much for your suggestions to our paper. These help to improve the manuscript. Following changes are done:
1) Please avoid grouped citation such as [5-9]. Please write one/two sentences describing their contribution to the literature.
Line 65-67: “Several studies have been carried out to establish optimal process conditions concerning surface quality and geometry, e.g. by analyzing the influence of dry process conditions [5], feed per stroke values [6], radial feed per stroke values during plunge rotary swaging [7] or true strain [8].”
2) Please mark clearly in figure 1: which is the axial direction and which is the tangential direction.
Figure 1: “Axial feed motion of the workpiece”, “Radial oscillating motion of the die”
Figure 4. Line 174-175: ”Sketch of the residual stress measurement positions (mm) at the outer surface of swaged tubes (z - axial residual stresses; x–tangential residual stresses).”
3) Page 9: line 264: Comparable microstructure was also found for the round and the flat rotary swaging setup in the present study. Please provide evidence for this statement.
Line 278. Additionally Figure 13.
4) Figure 7 and 8: Why the axial residual stress at the surface in figure 8 is different than the same positions in figure 7? It is a very crucial point.
Line 228-230: “To further investigate this fluctuations, measurements were carried out at several positions below the surface of a different R0500 sample, sample 2, which showed similar fluctuating residual stresses as observed at R0500 sample 1 with different magnitudes, see Fig.8(a)”
5) Figure 12: What is the depth of the residual stress measurement points?
The scale is corrected: from the surface to 480 μm.
Round 2
Reviewer 4 Report
The manuscript is good to go.